# Language Models Might Not Understand You: Evaluating Theory of Mind via Story Prompting

## Abstract

We introduce `StorySim`, a programmable framework for synthetically generating stories to evaluate the *theory of mind* (ToM) and *world modeling* (WM) capabilities of large language models (LLMs). Unlike prior benchmarks that may suffer from contamination in pretraining data, `StorySim` produces novel, compositional story prompts anchored by a highly controllable STORYBOARD, enabling precise manipulation of character perspectives and events. We use this framework to design first- and second-order ToM tasks alongside WM tasks that control for the ability to track and model mental states. Our experiments across a suite of state-of-the-art LLMs reveal that most models perform better on WM tasks than ToM tasks, and that models tend to perform better reasoning with humans compared to inanimate objects. Additionally, our framework enabled us to find evidence of heuristic behavior such as recency bias and an over-reliance on earlier events in the story. All code for generating data and evaluations is freely available.[1]

## 1 Introduction

Large language models (LLMs), especially models meant for reasoning, have shown increasing improvement on benchmarks that once had language models stumped. Contemporary models have made progress in reasoning through math problems, coding, and pure logical deduction (Jaech et al., 2024; Guo et al., 2025; Xia et al., 2025). However, as the field shifts towards improving reasoning abilities, we look to LLM's abilities to reason about mental states and the perspectives of other intelligent agents, a concept known as *theory of mind* (ToM). This has become an increasingly important ability, as an LLM without a sound ToM might not be able to understand the perspective of a user with harmful intent, or consider the effect its decisions could have on other people, who have their own goals and wishes. Reasoning over human perspectives requires understanding the world that the perspective is seated in; for this reason, we consider ToM as an extension of *world modeling* (WM), which is the broad ability of an LLM to internally represent the state of the world. An intelligent agent with advanced world modeling capabilities would naturally have advanced theory of mind, as well, since other agents (and their mental states) exist in the same world as the agent. For this reason we propose a flexible synthetic dataset that allows us to test both capabilities, which we call `StorySim`.

ToM is an ability that is measurable in children, where we test their ability to understand that other people have their own mental representations of the world that differs from their own. The prototypical ToM test is known as the *Sally-Anne* test (Baron-Cohen et al., 1985), where children are given a story akin to the following:

*Sally and Anne walk into a bedroom. Sally places a toy on top of the bed and leaves. Anne takes the toy and hides it under the bed. When Sally comes back into the room, where is the first place she will look for the toy?*

It may seem obvious to adults that the first place Sally will look is on top of the bed. However, it's been shown that children under the age of 4.5 think that Sally will look under the bed (Memisevic et al., 2018). This is hypothesized to be due to the fact that they have not yet developed ToM, so they cannot understand the fact that Sally and Anne have a different understanding of the toy's location.

---

[1]The code for and data generated by `StorySim` is available here.

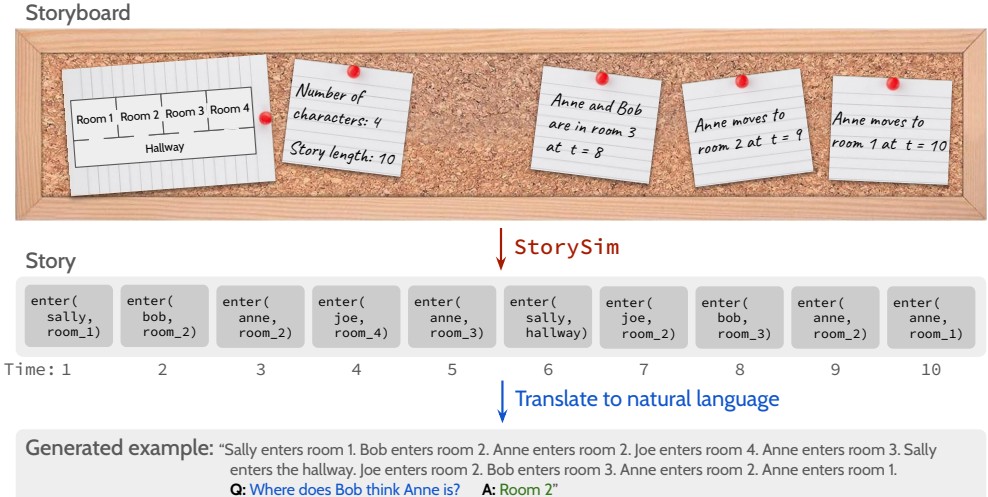

Figure 1: **(top)** Example STORYBOARD containing the graph of locations, the set of characters, requested story length, as well as a set of key events. **(middle)** Example `StorySim` generated-story using the given STORYBOARD. Each character can only move to locations corresponding to adjacent vertices in the graph (all characters start in the hallway). **(bottom)** The output of `StorySim` translated into a natural language QA example.

LLMs have been evaluated on problems of this nature, and initially showed success (Kosinski, 2023). However, small perturbations of this problem often confounds LLMs, indicating that this test has contaminated pretraining data, and LLMs may have overfit to this specific kind of problem, and therefore, they may not necessarily have a sound ToM mechanism (Ullman, 2023). In order to measure ToM ability in a way that is robust against data contamination, we have developed a benchmark that is synthetic, which guarantees that each example is novel and unseen during pretraining.

We propose `StorySim`, a framework that allows us to generate stories to evaluate ToM and WM abilities of language models. The strength of this framework is its high degree of programmability and flexibility, which allows us to generate stories with specific events that happen with respect to the perspectives of the characters in the story, which we refer to as the STORYBOARD. Aside from the STORYBOARD, the rest of the story is randomly generated, but guaranteed to be cohesive and consistent. An example of this framework is shown in Figure 1.

By leveraging `StorySim` to systematically evaluate current LLMs, we are able to address the following research questions in this paper:

**RQ1** Can we measure ToM abilities of LLMs in a way that is robust to data contamination and heuristics?

**RQ2** To what extent do LLMs rely on memorization in pretraining and heuristics to answer ToM questions?

**RQ3** Can ToM abilities in LLMs be evaluated independently of their general world modeling capabilities? Relatedly, do LLMs show stronger performance on general world modeling tasks or on ToM-specific reasoning?

All evaluations and code for generating using `StorySim` will be freely available at `[REDACTED]`.

## 2 Related Work

**Theory of Mind**: Existing research benchmarking ToM in language models focuses on prompting with stories followed by question-answering (Grant et al., 2017; Sap et al., 2019), including the TOMI dataset (Le et al., 2019). However, these are outdated, and have very likely leaked into the pretraining data of contemporary LLMs. In fact, Amirizaniani et al. (2024) demonstrates that trivial alterations of the TOMI dataset causes model performance to drop. In order to address data leakage,

newer benchmarks have been created (He et al., 2023; Chen et al., 2024b). Specifically, we position our work as similar to Sclar et al. (2024), where the authors present EXPLORETOM, a framework that allows for graph-based synthetic story generation and question answering. As in our work, this ensures that generated examples are relatively unseen by LLMs. However, `StorySim` is able to generate stories with more diverse settings, characters, and actions, and the use of non-determinism allows us to quickly and easily create a wide breadth of scenarios for LLMs to consider with minimal labor. Additionally, the high degree of programmability and control of `StorySim` work allows us to target and test for specific heuristics, as shown in Section 3, which existing benchmarks do not readily provide.

**World Modeling**: Though there is much work evaluating aspects of WM abilities of LLMs (Kočiský et al., 2018; Qiu et al., 2025; Qian et al., 2025), there are none, to the best of our knowledge, that examine the relationship between WM and ToM ability. We consider ToM to be an extension of world modeling since ToM requires the explicit modeling of mental states and how they may change over time, and mental states are dependent on the goals, limitations, and perspectives of agents. In the stories we generate, the character's perspectives only consist of what they observe around them, so to reason over this requires understanding the world in which they live and the limitations of each character's perception and knowledge. Wang et al. (2024) train a narrator and a role player model to interact within role-playing game environments. This setup implicitly requires both models to construct and maintain an understanding of the environment (WM), while the narrator must also infer the role player's perspective to guide narrative choices (ToM). However, the task itself does not directly evaluate either ToM or world modeling capabilities in isolation.

**Synthetic Datasets**: Though there are existing synthetic datasets meant for benchmarking LLMs (Patel et al., 2024; Shvets, 2025; Sood, 2024; Mishra & Prabakeran, 2025), there are none that evaluate ToM and WM abilities as we do. Synthetic datasets have been shown to be an effective method for training LLMs to perform well on logic tasks (Xie et al., 2025), clinical text classification (Tang et al., 2023), deductive reasoning (Saparov et al., 2023), and more. The diversity of synthetic data is an important factor in the evaluation of performance on both in- and out-of-distribution data (Chen et al., 2024a; Havrilla et al., 2024). For this reason, we developed `StorySim` to be highly programmable, allowing for a wide range of possible stories to be generated.

## 3 Methods

Broadly, the `StorySim` framework works by generating each story in two steps: (1) we create a STO-RYBOARD, which contains high-level information about the story we want to generate, and (2) using this STORYBOARD, `StorySim` generates a *story* that meets the specifications of the STORYBOARD and randomly generates sequences of events that fill in the gaps coherently and consistently.

In this dataset, we define a STORYBOARD as

$$D = (C, \phi, G, E, n)$$

where $C$ is a set of characters, $G$ is a directed graph describing locations in which the story happens, $E$ is a programmed set of events that occur in the story, and $n$ is the length of the story in terms of the number of events. The `StorySim` framework then generates a sequence of $n$ events, where each event can be written in the form $\phi(c, l)$ where $\phi$ is the action, $c \in C$ is the character performing the action, $l \in V(G)$ is the location of the action, and $V(G)$ are the vertices of $G$. These locations of the events follow the structure of $G$, so characters can only move between adjacent vertices in $G$. The only parts of the story that are deterministic are the events specified in $E$, which dictates specific time steps at which characters must perform specific actions in the story. Aside from this, all other events are randomly-generated, but are guaranteed to be cohesive with respect to the events specified in $E$. An example of such a story is provided in Figure 1. This sequence of events can be simply translated into natural language using templates, which maps each $\phi(c, l)$ into plain text.

Though a simple framework, `StorySim` can be used to create a wide variety of stories. While the generated stories are not the most complex in terms of prose or grammatical complexity, we can easily add different factors to stories that allow us to ask targeted questions to evaluate LLMs. Figure 2 is an example of a more complex story that offers the ability to ask many different questions (for example, how might Alice feel during her phone call?). By using events described in simple sentences, we minimize confounding factors related to sentence interpretation, ensuring that any model errors stem from limitations in the targeted ability.

In our experiments, we create many such STORYBOARDS that allow us to create stories with questions that can only be answered by models with sufficient ToM and WM abilities. The exact parameters of $D$ in our experiments are specified in Appendix A. We generate stories involving characters moving between a set of rooms connected by a hallway. We conducted experiments using different types of settings and found no significant difference in performance, as shown in Figure 9.

In order to test ToM abilities, we generate two types of ToM problems:

*First-order ToM problem*: Ask about one character's knowledge of another character's location. E.g.: "Where does Bob think Alice is?"

*Second-order ToM problem*: Ask about one character's knowledge of another character's knowledge of a third character's location. E.g.: "Where does Bob think Charlie thinks Alice is?"

To create stories that allow us to ask questions like the ones above, we create two types of STORYBOARDS which specify events for characters $S_1$, $S_2$, and $T$. In the above examples, $S_1$ is Bob, $S_2$ is Charlie, and $T$ is Alice. We use `StorySim` to generate stories to test first-orer ToM using the following STORYBOARD: $S_1$ and $T$ move to the same location at some time, $T$ moves to a different location. Then, unbeknownst to $S_1$, $T$ moves to a third location, where they remain for the entirety of the story. We also generate the second-order ToM stories using the following STORYBOARD: where $S_1$, $S_2$, $T$ move to the same location, then $S_2$ and $T$ later meet again at a different location, then $T$ moves to a different location on their own.

*RQ1 & RQ3*: In order to answer these research questions, we aim to generate stories that allow us to measure ToM and WM abilities in controlled experiments, while minimizing the chance that the model can exploit memorization from pre-training data. Using the second-order STORYBOARD as described above, we generate a large number of stories, and prompt the LLMs with the following three questions:

`ToM Prompt:` Where does $S_1$ think $S_2$ thinks $T$ is?

`WM-Human Prompt:` When $S_1$ and $S_2$ were in the same room as $T$, where did $T$ go?

`WM-Inanimate Prompt:` When $S_1$ and $S_2$ were in the same room as $T$, where was $T$ moved to?

The first question type requires the LLM to understand the perspective of $S_1$ and their perceived location of $S_2$ and $T$, thereby directly testing its ToM capabilities. The second question type is equivalent to the first, but instead of asking the model to consider the perspective of $S_1$, we prompt it to algorithmically search through the story to find an event where $S_1$'s perception of the locations of $S_2$ and $T$ changed. Thus, the second question is more a test of the model's world modeling ability, since we do not ask about the mental state of any character, and answering the question requires understanding of the events in the story. Finally, the third question type is equivalent to the second, but the story is edited to use inanimate objects instead of human subjects. This question is meant to test whether the model's world modeling ability is sensitive to whether the entities in the world are animate or inanimate, which shows us the difference in ability when the model is considering humans compared to anything else. The results of evaluating a suite of LLMs on these three prompts is shown in Section 4.2.

*RQ2*: One advantage of the design of `StorySim` is that it allows us to test for specific heuristics we suspect that models are exploiting to answer questions. We take recency bias as an example of a heuristic that LLMs may use when making predictions. The flexibility of `StorySim` enables us to easily setup an experiment to test this. Given a STORYBOARD for a first-order or second-order ToM problem, we ask whether the number of time steps between the event where $S_1$ last perceives $T$'s location and the event where $T$ makes their third and final move is predictive of model performance. Note that when $T$ makes their final move in the story, $S_1$'s earlier knowledge of $T$'s location becomes inaccurate. But since $T$'s last movement is a more recent event in the story, LLMs may utilize a heuristic where they predict $T$'s last location, despite the fact that $S_1$ did not see this movement. We refer to this number of time steps between these two events as the *mislead distance*. Our initial hypothesis was that increasing the mislead distance would increase the model's accuracy due to recency bias in the prompt. To test these heuristics, we create STORYBOARDS for both first- and second- order ToM problems, but we vary the mislead distance The results in Section 4.3, we evaluate multiple LLMs on these examples and evaluate their performance.

| Task | GPT-4 | Deepseek-R1 |
|---|---|---|
| Sally-Anne | 100% | 100% |
| Random Story | 75% | 87% |

Table 1: Accuracy on Sally-Anne vs the other ToM tasks. In the top row, models are prompted with the exact same Sally-Anne setup shown in the Introduction, in a story that's 5 events long. In the bottom-row, models are given a randomly-generated first-order ToM story of the same length.

In addition, we explore whether and to what extent models have overfit to simpler ToM questions by significantly increasing the number of subjects in the stories. We hypothesize that models may have seen question-answering prompts based on stories during pretraining, but it is unlikely they have such examples where the stories contain large amounts of characters. Being able to do so would imply that the models can generalize more robustly to stories with many mental states, and would therefore provide evidence of more robust/generalizable ToM and WM capabilities. We prompt the models using the `ToM Prompt` and `WM-Human Prompt`, shown earlier in this section, except we ask a first-order ToM question. The results of evaluating the performance of multiple LLMs using this story setup and two prompts are shown in Figure 8 and further discussed in Section 4.4.

> Charlie arrives at the field. Bob heads to the field. Bob goes to the store. Bob heads to the field Charlie goes to the store. Alice heads to the store. Alice and Charlie exchange phone numbers. Bob goes to the store. Alice goes to the cafe. Bob goes to the field. Charlie arrives at the cafe. Alice calls their friends on the phone to tell them what happened.

Figure 2: A story about characters meeting and exchanging phone number using the actions *arrives*, *heads*, *goes to*, and *heads to*. Each sentence is a generated event.

## 4 Results

### 4.1 Sally-Anne Questions Are No Longer Good Tests for ToM

To highlight the inadequacy of benchmarks that rely on the Sally-Anne problem to measure ToM, we used `StorySim` to mirror the Sally-Anne experiment using GPT-4 and DeepSeek-R1, using a STORYBOARD that specifies a first-order ToM problem with 5 events matching the experiment exactly. As shown in Table 1, both of these models had perfect scores on this task, which is far from the results of prompting the models using randomly generated stories of the same length.

### 4.2 Evaluating ToM vs WM Capabilities

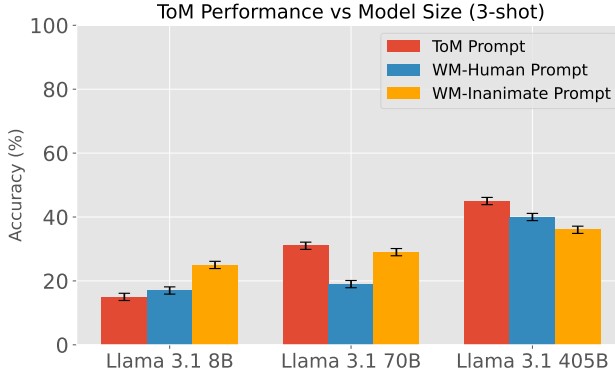

Figure 4: Performance by the Llama 3.1 Instruct family of models, using 3-shot examples given the ToM prompt, WM-Human prompt, and WM-Inanimate prompt. Similar to Figure 3, these stories have 100 events each.

We tested a suite of LLMs using the prompts discussed in Section 3. Starting with a STORYBOARD describing a second-order ToM event, we generate 100 stories consisting of 100 events with which we query each model. Figure 3 shows the results of zero-shot prompting GPT-4, DeepSeek-R1, the Llama 3.1 Instruct family, Llama 3.2 Instruct, Llama 3.3 Instruct, and Qwen QwQ (Achiam et al., 2023; Guo et al., 2025; Touvron et al., 2024; Yang et al., 2025). GPT-4, Llama 3.2, and DeepSeek-R1 are the only models that perform better on the ToM prompt than on the WM-Human prompt. While the remaining models score higher on WM-Human than on ToM, their raw accuracies on both

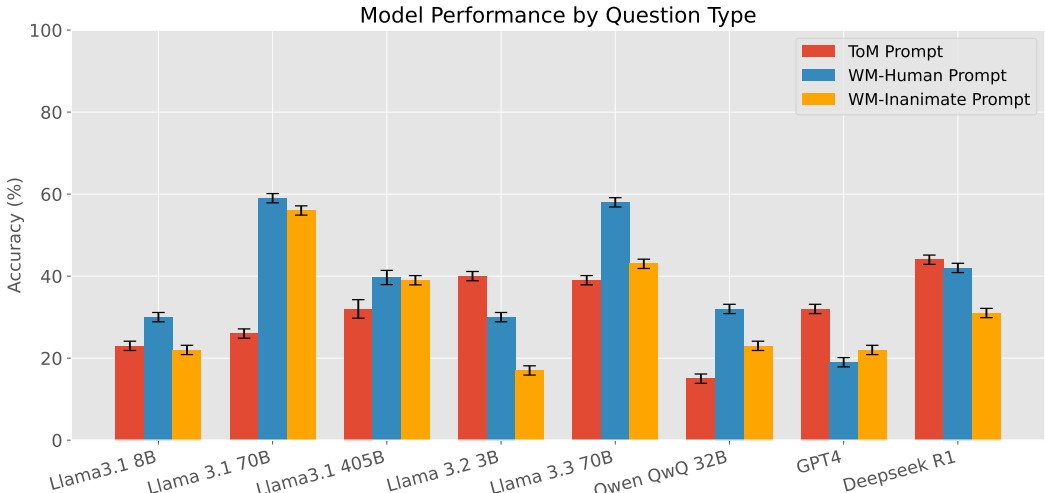

Figure 3: Model performance per model using the ToM, WM-Human, and WM-Inanimate prompts. For each of these prompts, the models are given a story containing 100 events, and we give 100 such stories to each model. The black bars represent the 95 percent confidence interval.

tasks are generally higher—or at least comparable—to those of Llama 3.2 and GPT-4. An exception is DeepSeek-R1, which achieves the highest ToM accuracy overall. In every model except GPT-4, the WM-Human score is higher than WM-Inanimate, indicating that considering human subjects makes the LLMs more accurate in tracking changes in location than with inanimate objects. However, the fact that most of the model's ToM scores are lower than WM scores leads us to believe that LLMs have greater difficulty when tasked to reason over the mental states of characters.

Since increasing model scale allows LLMs to perform better on many other tasks, we also endeavor to test whether or not ToM and WM abilities are enhanced with more parameters. To this end, we test three models of varying sizes in the Llama 3.1 family.

Looking at the Llama 3.1 models in Figure 3, the 70B model outperforms both the 405B and 8B models in both the WM tasks. However, we find that the ToM score increases as model size increases. In order to examine this further, we tested the Llama family of models in a 3-shot setting as well. Figure 4 shows a significant correlation between model size and both ToM and WM-Inanimate scores. WM-Human scores are highest in the 405B model as well. These results suggest that, within the Llama 3.1 architecture, there is a clear correlation between ToM and model size. In the 3-shot setting, this correlation is apparent in WM abilities as well.

### 4.3 Do LLMs Rely on Heuristics for ToM?

Our next experiments aim to address whether or not LLMs are exploiting simple heuristics to answer our questions rather than relying on an actual ToM mechanism. We used `StorySim` to create controlled randomly generated stories that, in tandem with us tracking the responses of the model, would allow us to easily determine whether there is a pattern in the responses the LLMs give us, indicating that they are relying on a shortcut to respond to the prompt.

#### 4.3.1 How Sensitive Are LLMs to Recency Bias in ToM?

In order to gain insight into whether or not LLMs rely on recency bias or a heuristic, specifically where the model predicts $T$'s last location without considering $S_1$'s mental state, we evaluated GPT-4 and DeepSeek-R1 on first- and second- order ToM problems while increasing the mislead distance (i.e., the number of steps after $S_1$ sees $T$ move to another location before $T$ moves to a different location altogether). The left plot in Figure 5 displays the results in the first-order setting, where DeepSeek-R1 performs better as the mislead distance increases, suggesting that the model can exploit recency bias in the prompt. If the perspective of $S_1$ was more explicitly represented in the model's

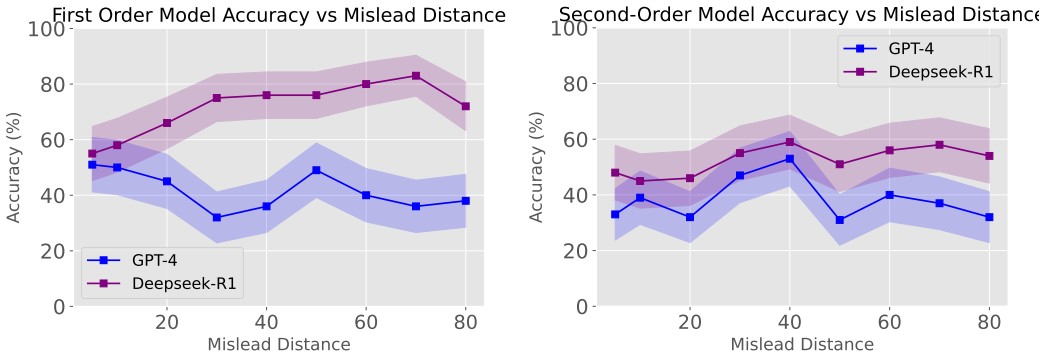

Figure 5: **(left)** Model accuracy on the first-order ToM task as a function of the mislead distance (i.e., the number of steps after $S_1$ sees $T$ move to another location and before $T$ moves to a different location altogether). **(right)** Model accuracy on the second-order ToM task as a function of mislead distance. Shading indicates a 95% confidence band.

internals, then the time step of $T$'s move would not affect the model's performance on the task. However, this is not the case. GPT-4 struggles with the task, as its accuracy is much lower than DeepSeek-R1 and there seems to be no apparent pattern between mislead distance and first-order ToM performance.

Figure 5 (right) shows the results of the second-order ToM version of the experiment. Interestingly, though DeepSeek-R1 still shows slight improvement as the mislead distance increases, the scores for both models appears erratic. For both models, a sudden spike in score occurs at mislead distances of 30 and 40, followed by a decrease at 50. However, DeepSeek-R1's improvement at higher mislead distance still points to a reliance on recency bias.

### 4.3.2 *What Kind of Mistakes do LLMs Make?*

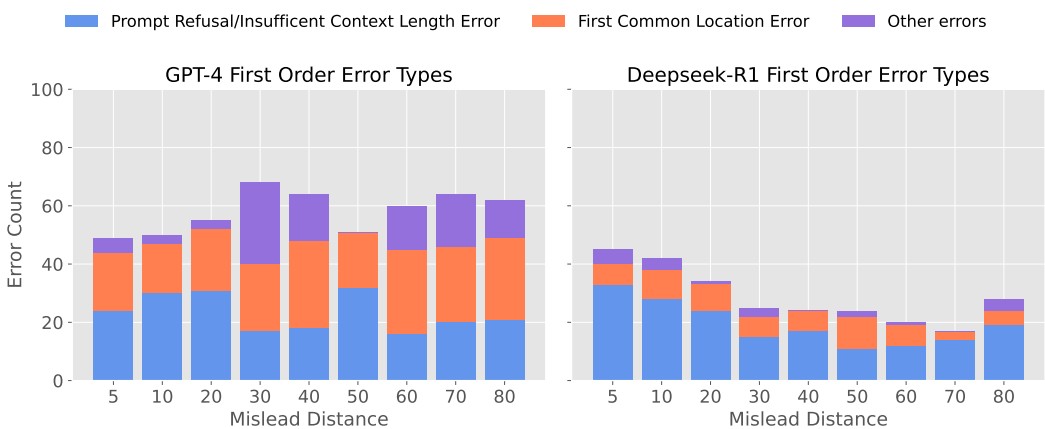

Figure 6: Number of mistaken examples (out of 100), with each example categorized by error type, for the first-order ToM mislead experiments on **(left)** GPT-4 and **(right)** DeepSeek-R1.

The most common error for GPT-4 is prompt refusal, while for DeepSeek-R1 it is insufficient context length. For DeepSeek-R1, this looks like the model trying to reason through an answer, but runs out of context length. For GPT-4, this looks like the model replying with, "not enough information". Aside from this, we also found that the most common error is to repeat the location of the first place where characters are in the same location. For example, in Figure 1, this would correspond to the LLMs guessing room_2, since that is where Bob and Anne cross paths on their way to room_3, as specified by the STORYBOARD. We refer to this as the *First Common Location Heuristic*. For both

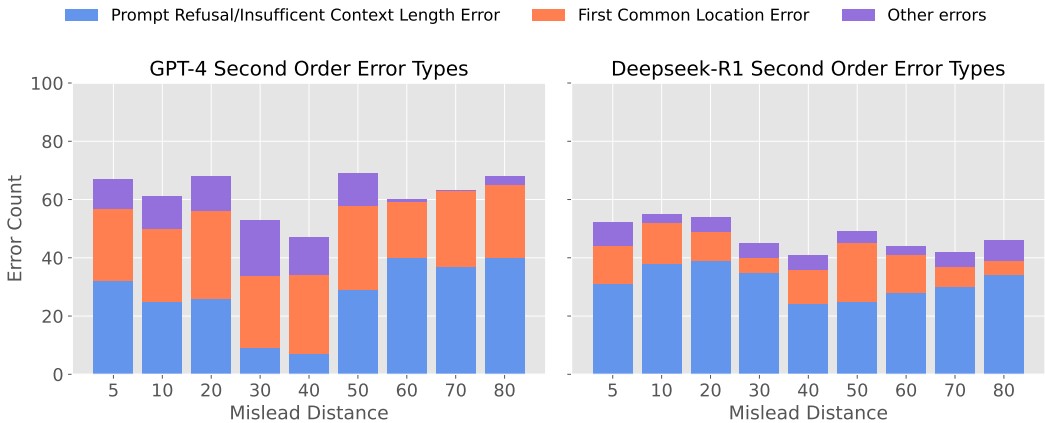

Figure 7: Number of mistaken examples (out of 100), with each example categorized by error type, for the second-order ToM mislead experiments on **(left)** GPT-4 and **(right)** DeepSeek-R1 out of 100 examples.

the first- and second-order ToM tasks, we want to quantify the relative proportions of each of the types of errors, so we tracked model responses and categorized them as we increased the mislead distance. The results of this are shown in Figures 6 and 7.

In the first-order experiments, DeepSeek-R1's performance improvement as mislead distance increases is largely due to a reduction in insufficient context length errors. This suggests that the model struggles to reason about the mental states of characters when the relevant ToM event is less recent in the story. As shown in Appendix C, these erroneous responses from DeepSeek-R1 often involve the model attempting to enumerate the story's events, trying to determine when $S_1$ last saw $T$, and then restarting its reasoning chain. When DeepSeek-R1 does produce a concrete but incorrect answer, it is most often a First Common Location Error, suggesting a tendency to memorize the first location where $S_1$ and $T$ are seen together.

GPT-4 exhibits a slight downward trend in prompt refusal as mislead distance increases, while its rate of First Common Location Heuristic remains roughly constant. However, GPT-4 predicts locations other than the initial crossing point far more frequently

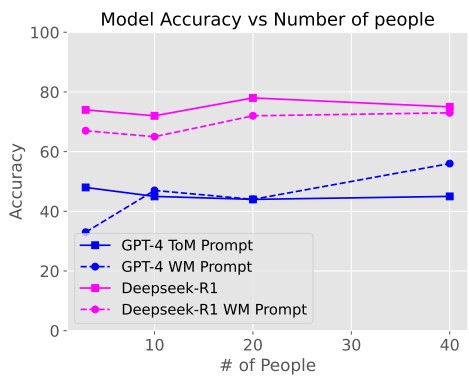

Figure 8: Model accuracy vs the number of characters in the story. Solid lines show the accuracy of the ToM prompt responses, dashed lines show the accuracy of the WM prompt responses.

than DeepSeek-R1, which ultimately leads to worse performance on the overall task. Although it frequently falls victim to First Common Location Heuristic, GPT-4's responses are considerably more variable than those of DeepSeek-R1.

In the second-order experiments, both models show sharp performance improvements at mislead distances of 30 and 40, which appear to be driven by reduced rates prompt refusal and running out of context length. The second-order experiments do not clearly indicate reliance on a specific heuristic by DeepSeek-R1 model; rather, the task appears sufficiently challenging to induce complete model failure in many cases. However, GPT-4 does respond with the first common location in many cases, indicating a reliance on the First Common Location Heuristic.

## 4.4 Do More Characters Increase The Difficulty of ToM?

In an effort to determine whether or not LLMs rely on having only seen ToM questions about stories with only a *few* characters, we experimented with increasing the number of characters in the stories. By keeping the length of the story at 100 events, of which only a handful are specified by the

storyboard, increasing the number of characters yields stories where an increasing proportion of the events features a novel character.

In the first-order ToM prompt, the number of characters does not appear to have a statistically significant effect on ToM performance for either DeepSeek-R1 or GPT-4. However, when prompted with the WM equivalent of the first-order ToM task, GPT-4 struggles with fewer characters but improves markedly as the number of characters increases. This suggests that the presence of more distinct human entities in the prompt elicits more accurate world modeling behavior in GPT-4, enabling it to respond more accurately to our prompts.

# 5  Conclusion and Future Work

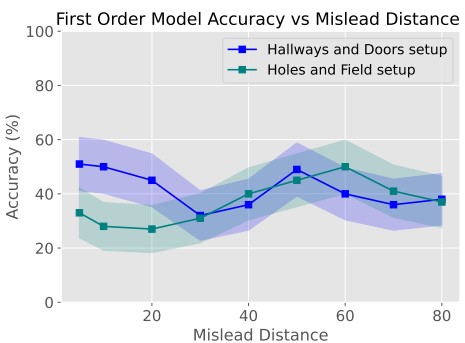

Figure 9: Model accuracy vs mislead distance using GPT-4 using two different STO-RYBOARDS: one that features characters entering and exiting rooms from a hallway, and one that has characters in a field jumping in and out of holes in the ground. Otherwise, the stories are generated in the same way.

We present `StorySim`, a framework that synthetically generates random stories from a STORYBOARD, a high-level outline containing a number of key events. This framework enables us to evaluate the ToM and WM abilities of LLMs in a fashion that is robust against data contamination, and to investigate whether world modeling ability is a good predictor of ToM ability. It also allows us to identify heuristics that LLMs may use in place of robust ToM reasoning.

Our results show that most LLMs exhibit generally stronger world modeling ability than ToM. However, we also observe promising signs that increasing model size may improve ToM performance. Our error analysis reveals that LLMs often memorize incorrect character locations within the stories. Additionally, targeted story generation to expose heuristic behavior suggests that LLMs are more likely to answer ToM questions correctly when changes in a character's mental state occur closer to the end of the prompt.

We hope this work inspires future efforts to benchmark ToM and social reasoning abilities in LLMs. While we currently focus on each character's perception of each other's locations, `StorySim`'s programmability allows for more complex stories than those presented here, including richer actions, environmental interactions, goal-directed behavior, and social attitudes and other broader aspects of mental state reasoning. As LLMs are increasingly used in socially-sensitive domains—such as resume filtering, college admissions, and healthcare—a rigorous benchmark of their ToM and social reasoning capabilities is essential. `StorySim` represents a step toward understanding how LLM's limitations in ToM affect their applicability in real-world settings, as well as the potentially deleterious consequences of their wider deployment.

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

## A   Further Experiment Details

The storyboard $D$ used in the majority of our experiments was created using the following:

**Mislead experiments**:

- Characters $C$ = {Alice, Bob, Charlie, Danny, Edward, Frank, Georgia, Hank}
- Actions $\phi$ := {enters}
- Location graph $G$ which has six locations, where each vertex has exactly three outgoing edges, representing three locations that can be accessed from that location.

- Manually specified events $E = [\{$name: cross_paths,actors: $\{S_1, T\}$, location: location_1,t=10$\}$,
  $\{$name:move, actor:$T$, location: second_location , t=11 $\}$,
  $\{$name: exclusive_random, actors: $\{S_1, T\}$, t=12, stop: $12 + mislead\_distance$ $\}$,
  $\{$ name: move, actor:$T$,location:third_location, t= $12 + mislead\_distance$ $\}$
  $\{$ name: exclusive_random, actors: $\{S\_1, T\}$, stop: $n$ $\}$ ]
- $n = 100$

The "exclusive_random" event makes StorySim randomly generate sequences that don't affect the actors specified. Note that for the second-order experiments, the only difference is the introduction of $S_2$, and an additional cross paths event is added to create the second layer of ToM. In every trial, $S_1$, $S_2$, and $T$ are randomly picked from $C$. Further, $location\_1$, $second\_location$, and $third\_location$ are all randomly generated from $G$.

The storyboard used for the experiments described in Section 4.2 is the same as the second-order mislead distance experiment, but the mislead distance is fixed at 30.

## B  Narrative Variations

While the generated stories are not the most complex in terms of the prose they use or grammatical complexity, we can easily add different factors to stories that allow us to ask specific questions. We provide Figure 10 as a supplementary example of this. If one wishes to make more detailed versions of the stories generated by our framework, it's possible to use an LLM to follow the generated events and generate a polished story. The story shown in Figure 11 is the result of asking GPT-4o to do such a thing (Hurst et al., 2024). Though not shown here, StorySim can generate any story one can think if it can be expressed as a series of actions between an actor and another entity or location. Our examples only look at stories where the actions are actors interacting with their environment. However, recalling the notation from Section 3, imagine a story generated where $\phi := whisper$ and $G$ was an adjacency graph specifying a social network of characters who can only whisper to other specified characters. Further, imagine creating multiple StorySim objects and creating a story by sampling randomly generated events from both of them. This could be used to create stories that target characteristics with multiple dimensions. We invite readers to use StorySim to create custom stories that may fit their needs.

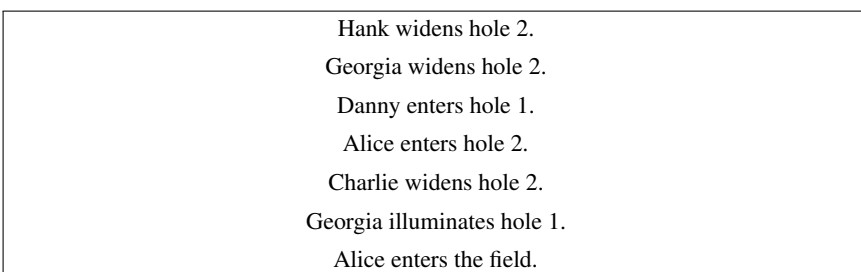

Figure 10: StorySim story generated with three relations: *enters*, *illuminates*, and *widens*

> Charlie arrived at the field first, scanning the open space with an expectant glance. Moments later, Bob made his way there too, walking at a steady pace. Whatever brief conversation they might have had was short-lived—Bob soon departed for the store.
> His errand complete, Bob returned to the field once more, retracing his earlier steps. Charlie, now done with his own time at the field, headed off toward the store.
> Elsewhere, Alice also made her way to the store. It happened that she and Charlie were there at the same time. After a short conversation near the entrance, they exchanged phone numbers before parting ways.
> Bob stopped by the store again—not his first visit that day—perhaps tying up a loose end. Meanwhile, Alice took a detour and went to the café. It was quieter there, a place to think or simply pause.
> Not long after, Bob returned yet again to the field, following the same path he'd already walked earlier. Charlie, too, arrived at the café—perhaps by coincidence, or perhaps not.
> Some time passed. Then, with her phone in hand, Alice made a call. One by one, she reached out to their friends to tell them what had happened.

Figure 11: Using events from Figure 2 to create a more interesting story by asking an LLM(GPT-4o) to use the structure of the generated events to generate a natural sounding story, without adding any new events that deviate from the original.

## C  Example of DeepSeek-R1 Insufficient Context Length

Below is an example of one of DeepSeek-R1's responses where it runs out of context length. Note that all errors that it reports finding in the story are untrue. The ellipsis indicates truncation of the prompt for brevity, but where the sentence cuts off in the last step is not truncation, that is where it ran out of context length.

```
<think>
Okay, let's try to figure out where Alice and Bob were last in the
same room and where Alice saw Bob go when he left.
The question is a bit tricky because I need to track both
Alice and Bob's movements through all the
rooms and see when they were last together.
Then, once they part ways, I need to note where Bob
went next from Alice's perspective.

First, I'll start by going through the story step
by step, noting each time Alice or Bob moves. Since all characters
start in the hallway, their initial positions are there. Let's list out their
movements in order:

1. Edward enters room_1.
2. Alice enters room_1. So now Alice is in room_1 with Edward.
...
35. Georgia enters room_5. Georgia was in room_5, but maybe she left?
Wait, step 23: Georgia was in room_5. Then step 24: Frank leaves room_5.
Then step 25: Charlie enters room_2. Then step 26: Charlie to hallway.
Step 27: Frank to room_4. Step 28: Danny to room_2. Step 29: Frank to room_3.
So Georgia was in room_5 until step 35: Georgia enters room_5 again? Wait,
maybe she was in room_1 before. Let me check. Step 15: Georgia enters room_1.
Then step 23: Georgia enters room_5. So she was in room_1, then room_5.
Then step 35: Georgia enters room_5 again? Wait, maybe that's a typo.
Let me check the original movements again.

Looking back at the original list:

After step 15: Georgia enters room_1.

Then step 23: Georgia enters room_5. So she moves from room_1 to room_5.
```

```
...
Step 34: Frank enters room_2.

Step 35: Georgia enters room_5. Wait, but she was already in room_5 at
step 23. So maybe she left room_5 at some point? Let me check.
The steps after step 23:

After step 23 (Georgia enters room_5), the next movement
involving Georgia is step 35: Georgia enters room_5 again.
But unless she left room_5 in between, she would still be there.
But according to the given movements, characters only move when
specified. So if Georgia entered room_5 at step 23, she stays
there until she moves again. The next move for Georgia is
step 35: Georgia enters room_5 again. Wait, that's
redundant. Maybe it's a mistake, or perhaps she was in
another room. Let me check the steps again.

Wait, step 23: Georgia enters room_5. Then step 24: Frank leaves room_5.
Then step 25: Charlie enters room_2. Step 26: Charlie to hallway.
Step 27: Frank to room_4. Step 28: Danny to room_2.
Step 29: Frank to room_3. Step 30: Frank to room_4.
Step 31: Frank to room_3. Step 32: Frank to room_2.
Step 33: Frank to room_1. Step 34: Frank to room_2.

Georgia's movements:

- Enters room_4 (step 8)
- Enters room_1 (step 15)
- Enters room_5 (step 23)
- Then step 35: Georgia enters room_5 again.
Maybe she was in room_1 again? Wait, no. After step 15, she was in room_1.
Then step 23, she moves to room_5. Then step 35: she's entering room_5 again.
That doesn't make sense unless she left. But according to the given steps,
she didn't leave. So perhaps this is a mistake,
but I'll proceed as per the given data.
Maybe she was in room_5 and then enters again,
which doesn't change her location. So she's still in room_5.

Continuing:

35. Georgia enters room_5. (Still there)
...
53. Danny enters room_1. (From room_4 to room_1)
54. Georgia enters
```

