# OpenReview forum: "Language Models Might Not Understand You: Evaluating Theory of Mind via Story Prompting"
_colmweb.org/COLM/2025/Workshop/Social_Sim — Social Sim'25_

### Official Review · Reviewer_TNtY · 2025-07-16
**Structured ToM Evaluation**

**Rating:** 7
**Overall Assessment:** 4
**Confidence:** 3

**Review:**

# Pros:
1. **Well-structured and controllable evaluation setting**

    The STORYBOARD mechanism enables precise control over visibility, temporal ordering, and event structure. By explicitly defining these constraints, the framework supports the controlled generation of belief reasoning scenarios with varied information access and timing.

2. **Clear separation of reasoning types**

    The comparison between ToM, WM-Human, and WM-Inanimate questions is well designed to isolate belief reasoning from observational recall. This allows the same story context to support contrasting inference targets with minimal confounding.

3. **Targeted perturbation variables to examine heuristics**

    The introduction of factors like mislead distance allows the authors to test whether models default to recent observations or maintain belief-based state representations.

4. **Potentially extensible story generation pipeline**

    StorySim’s modular story construction can support more complex ToM tasks in future work, such as deception, multi-agent belief conflict, or intent inference, using the same structured generation engine.
# Cons:
1. **Belief tracking is evaluated only through final answers**

    The current setup does not expose intermediate model behavior, such as belief state updates or perspective-specific memory. As a result, it remains difficult to determine whether correct answers stem from structured belief modeling or superficial correlation patterns.


2. **Social reasoning is limited to visibility and location**

    Despite the flexibility of the story generation pipeline, the current tasks focus primarily on physical location and visibility constraints. The benchmark does not explore more socially nuanced belief states, such as understanding false intent, deception, or emotional misalignment.

**Comments Suggestions And Typos:**

**Evaluation focuses on single-question judgments per story**

    While the contrast between ToM and WM already reveals meaningful behavioral differences, adding questions that probe the same belief from multiple angles would be beneficial. Questions involving action justification, communication simulation, or counterfactual reasoning could help determine whether the model maintains a coherent, perspective-sensitive representation rather than simply generating isolated, task-specific responses.

**Paper Summary:**

This paper introduces **StorySim**, a synthetic benchmark designed to evaluate the Theory of Mind (ToM) capabilities of large language models (LLMs). The system uses a STORYBOARD abstraction that defines structured, programmatic stories involving multiple characters, visibility conditions, and spatial locations. Each story is used to generate a single question targeting one of three reasoning types: **ToM**, **World Modeling for humans (WM-Human)**, or **World Modeling for inanimate objects (WM-Inanimate)**. The authors use this controlled setup to compare the performance of various LLMs (e.g., GPT-4, DeepSeek, LLaMA2) and introduce perturbations such as **mislead distance** to test whether models are relying on surface heuristics like recency. Results show a consistent performance gap between WM and ToM questions, suggesting that current models may not reliably construct or maintain perspective-specific belief representations.

**Relevance:**

4

**Summary Of Strengths:**

1. **Well-structured and controllable evaluation setting**

2. **Clear separation of reasoning types**

3. **Targeted perturbation variables to examine heuristics**


4. **Potentially extensible story generation pipeline**

**Summary Of Weaknesses:**

1. **Belief tracking is evaluated only through final answers**

2. **Social reasoning is limited to visibility and location**

---

### Official Review · Reviewer_k6NB · 2025-07-20

**Rating:** 7
**Overall Assessment:** 4
**Confidence:** 4

**Review:**

This is an ambitious and well-constructed study that presents a thoughtful and layered benchmark for testing ToM in machine learning systems. The three research questions guide the reader through increasing cognitive depth, and the authors do a commendable job identifying flaws in prior benchmarks. However, several interpretation points lack clarity. For example, performance gains at greater misleading distances appear to contradict the authors’ claim that models struggle with less recent ToM events. Furthermore, the rationale for certain methodological decisions — such as movement restriction to adjacent nodes — could benefit from clearer justification.

The treatment of world models (WM) as a foundation for ToM is compelling, though it remains somewhat ambiguous whether ToM is modeled as an emergent property or a distinct reasoning module. While the results are generally well presented, the lack of chance-level benchmarks across figures and inconsistent error attribution (e.g., “running out of context length”) slightly dampens confidence in the conclusions.

**Comments Suggestions And Typos:**

Consider adding chance-level performance lines to all figures for better interpretability.

Clarify why adjacent-only movement is enforced; if it's due to graph topology, explain its implications.

Include a simple example illustrating how Amirizaniani (2024)’s changes affect performance.

Expand on how world models integrate or differentiate from explicit ToM reasoning.

Re-express assumptions about DeepSeek’s recency bias with more quantitative backing.

Investigate whether prompt refusal/context errors arise from format/design issues rather than context length overflow.

For Figures 6 & 7, explore prompt manipulation strategies to reduce refusal or insufficient context errors.

Strengthen explanation regarding animate object recognition vs. mental state tracking in Section 4.2.

**Ethical Concerns:**

No overt ethical concerns were identified.

**Paper Summary:**

The paper introduces a novel benchmark for evaluating Theory of Mind (ToM) reasoning in language models, with tasks designed to assess both first- and second-order ToM capabilities. It identifies weaknesses in existing benchmarks due to potential data leakage and proposes an updated dataset aimed at more rigorous evaluation. The authors present three research questions of increasing complexity and analyze model performance across multiple axes — including animate object recognition, misleading event distances, and prompt refusals — using DeepSeek and GPT-4 among others.

**Relevance:**

4

**Summary Of Strengths:**

- The benchmark is robust, well-scaffolded, and introduces meaningful challenges beyond superficial behavior recognition.

- The layered complexity of the research questions encourages deeper cognitive modeling.

- The authors directly acknowledge and address data leakage in prior benchmarks.

- Integration of world model theory as a lens for ToM is conceptually innovative.

- Thorough experimental coverage and diverse model comparisons.

**Summary Of Weaknesses:**

- Interpretations surrounding misleading event distance and recency bias lack alignment with actual performance trends.

- Restriction of character movement to adjacent nodes may limit ecological validity; justification tied to graph-based structure should be clarified.

- Absence of chance-level performance baselines in figures reduces interpretability.

- Attribution of errors to context length may be inaccurate given the task size.
- Claims about DeepSeek’s distinctive bias (e.g., recency) require stronger evidence beyond curve parallelism with GPT-4.

---

### Meta-Review · Area_Chair_PFUk · 2025-07-21

**Recommendation:** Accept

**Metareview:**

--